# Gesture2Vec: Clustering Gestures using Representation Learning Methods for Co-speech Gesture Generation

## Abstract

Co-speech gestures are a principal component in conveying messages and enhancing interaction experiences between humans. Similarly, the co-speech gesture is a key ingredient in human-agent interaction including both virtual agents and robots. Existing machine learning approaches have yielded only marginal success in learning speech-to-motion at the frame level. Current methods generate repetitive gesture sequences that lack appropriateness with respect to the speech context. In this paper, we propose a Gesture2Vec model using representation learning methods to learn the relationship between semantic features and corresponding gestures. We propose a new conceptual framework that considers gestures as a non-verbal language itself. Our approach first converts gesture sequences into symbolic chunks, using frame and sequence level autoencoders and rigorous training techniques to learn the vocabulary. Using this higher-level representation, we then take advantage of a machine translation model to learn translations of text to discrete sequences of associated gesture chunks in the learned gesture space. Ultimately, we use these quantized gestures as input to the autoencoder's decoder to produce gesture sequences. The resulting gestures can be applied to both virtual agents and humanoid robots. Ablation studies support that our gesture chunking approach, fixed decoder weights, and vector quantization are the main drivers of diversity in objective gesture diversity measures. Further subjective and objective evaluations confirm the success of our approach in terms of appropriateness, human-likeness, and diversity.

## 1 Introduction

Non-verbal behaviour is an indispensable part of our daily communication. Prior research stated that $70-93\%$ of communication is non-verbal, including facial expression, hand gesture, body pose, and vocal tones (Mehrabian, 2017; Lapakko, 2007). People spontaneously gesticulate to complement verbal channels during the speech to convey messages (McNeill, 2011; De Ruiter et al., 2012; Cassell et al., 1999). Hence, integrating non-verbal communication skills into social robots and virtual agents is crucial for compelling interactions (Minato et al., 2004; Woods et al., 2004; Breazeal et al., 2005). Gestures originate from semantic features(Chu & Kita, 2016) and are characterized by speech context (Lücking et al., 2013). Therefore, co-speech gesture generation has been typically addressed using textual and acoustic features to generate relevant gestures.

Gesture synthesis can be categorized into deterministic and probabilistic models. Deterministic models predict a single output for a given input, while probabilistic models estimate a plausible output probability distribution conditioned on the given input. The limitation of deterministic data-driven methods is that they neglect to span the variation of many dimensions in data space (Fragki-adaki et al., 2015; Ferstl et al., 2019). Probabilistic models are potentially able to capture a broader gesture space, but probabilistic models also suffer from the posterior-collapse problem (Bowman et al., 2015) which results in repetitive gestures close to the average. The problem arises when the one-to-one mapping assumption disregards the one-to-many relationship between speech and gesture. This is a common problem among generative models for gestures (Yoon et al., 2019; Ginosar et al., 2019), as well as in other domains, e.g. image generation models that produce a limited set of blurry and similar images (Lucas et al., 2019a), and early attempts for natural language genera-

tion (Bowman et al., 2015; Kingma et al., 2016; He et al., 2019). Literature attempted to address this posterior collapse problem using different techniques such as adversarial training (Goodfellow et al., 2014; Ginosar et al., 2019; Arjovsky et al., 2017; Srivastava et al., 2017), variational autoencoders (Higgins et al., 2016; Mi et al., 2018; Ling et al., 2020), normalizing flow (Alexanderson et al., 2020), vector quantization (Oord et al., 2017) (Razavi et al., 2019), weakened decoders (Yang et al., 2017; Semeniuta et al., 2017). However, as reported, prior work (Yoon et al., 2019; Ferstl et al., 2019; Kucherenko et al., 2020b) were not successful in capturing long-term dependencies. Indeed, the model parameters focus on imperceptible and local features such as continuity among consecutive frames, most likely due to mode collapse.

Inspired by advances in other machine learning fields, especially natural language processing, we model gestures as a language that contains a vocabulary, and then perform text-to-gesture as a language translation task. Specifically, we integrate unsupervised representation learning methods and a machine translation algorithm. First, we reduce body pose dimensionality using a Denoising Autoencoder at the frame level. Then, we perform a discretized motion representation learning to cluster similar motion sequences to a symbol. This vector quantization method is a form of cluster pattern recognition where each motion sequence is assigned to a particular word from a codebook. Finally, we train a machine translation model to translate between utterances and their accompanying gestural motion symbols. This method is effective since it mitigates the complexity of gesture space and focuses on longer dependencies. In addition to traditional evaluation measurements, we also present new objective and subjective metrics to evaluate the diversity of gestures.

## 2 RELATED WORK

We first review co-speech gesture generation methods. Next, we discuss recent gesture generation approaches and their limitations. Finally, we introduce the deep neural network that we used and its advantages over prior work.

Early attempts for gesture generation use blending to smooth feasible motion clips selected from a database. Rule-based algorithms (Cassell et al., 2004; Huang & Mutlu, 2012), hidden Markov models (Levine et al., 2009), conditional random fields (Levine et al., 2010), and hybrid systems (Kipp, 2005; Neff et al., 2008) (Chiu et al., 2015) were used to select proper gestures conditioned on a given input. It is now recognized that these kind of systems require extensive efforts to annotate data and cannot generate gestures for unseen inputs. Also, they provide a single prediction for a given input and lack variation of generated movement. Furthermore, scheduling the gestures with speech is challenging since they may not be precisely synchronized (Butterworth & Hadar, 1989; Kendon, 2004) despite originating from the same source (McNeill, 2011).

Recent deep generative models, e.g. VAEs (Kingma & Welling, 2013; Higgins et al., 2016), GANs (Goodfellow et al., 2014; Abdal et al., 2019), and transformers (Vaswani et al., 2017; Brown et al., 2020; Dehghani et al., 2018), achieved notable results on different tasks. A generative model comprises the joint probability of given data and output. Therefore, it can generate new plausible instances by taking samples from the learned distribution (Hasegawa et al., 2018). Generative models have been used for human motion generation (Yan et al., 2018; Hernandez et al., 2019; Ling et al., 2020) as well as co-speech gesture generation (Ginosar et al., 2019; Yoon et al., 2019; Ferstl et al., 2021; Li et al., 2021). However, co-speech gesture generation is more challenging since the relationship between speech and motion is complex (Butterworth & Hadar, 1989).

Input to the generative system can be supplied from either speech-text, audio, or both modalities with a broad range of semantic and acoustic features (Kucherenko et al., 2021c). Systems that use acoustic features led to generated beat gestures according to the speech rhythm (Hasegawa et al., 2018; Ferstl et al., 2020; Ginosar et al., 2019; Kucherenko, 2018; Pouw & Dixon, 2019), and text-based systems (Yoon et al., 2019; Ishi et al., 2018) produced more semantically aware gestures. Although text-based models capture communicative features, they lack the strong effect of speech acoustics, i.e. intonation, prosody, and loudness, on expressed gestures (Pouw et al., 2020). Recent work benefiting from both modalities generated more compelling co-speech gestures in terms of appropriateness and naturalness (Yoon et al., 2020).

Typically, co-speech gesture generation systems build upon an encoder and decoder architecture (Hasegawa et al., 2018; Kucherenko et al., 2019; Yoon et al., 2019; Kucherenko et al., 2020a; Ferstl

& McDonnell, 2018). Recurrent Neural Networks (RNN) structures have been used extensively for the encoder and decoder (Hasegawa et al., 2018). However, RNN based models suffer from the error accumulation problem and are not good at capturing long-term human motion dependencies (Hernandez et al., 2019). While Convolutional Neural Network (CNN) based gesture generation models are not vulnerable to accumulation problems, CNNs are prone to regress to the average motion and generate repetitive gestures (Li et al., 2021). Variational Autoencoders were also proposed for the co-speech gesture generation task to generate more realistic and diverse gestures, e.g. (Rezende et al., 2014; Li et al., 2021; Kucherenko, 2018). VAEs with strong decoders tend to ignore the latent variable and learn the mode of the output data. This problem, known as "posterior collapse" causes the model to generate slightly similar outputs close to the average (Lucas et al., 2019b).

Literature considerably explored adversarial training to generate more realistic and diverse gestures (Ginosar et al., 2019; Ahuja et al., 2020; Yoon et al., 2020; Li et al., 2021; Ferstl et al., 2020; 2021). Although they brought attractive results, GANs are hard to train (Lucic et al., 2017) and suffer from mode collapse (Tulyakov et al., 2018). Normalizing-flow based models have GANs advantages while replacing adversarial loss by classical likelihood maximization training (Kingma & Dhariwal, 2018) and efficient probabilistic inference of VAEs (Henter et al., 2020) (Kingma & Dhariwal, 2018). Furthermore, autoregressive models on discrete data achieved impressive results in many sequences-to-sequence tasks such as machine translation (Wang et al., 2019), image generation(Salimans et al., 2017) (Razavi et al., 2019), and speech synthesis (Oord et al., 2016) (Gârbacea et al., 2019). VQ-VAE models learn a discrete latents space, enable us to use autoregressive models on the posterior, and does not suffer from "posterior collapse" (Oord et al., 2017). Ordinarily, gesture generation systems use a generator network that produces gestures from a latent code. To better regress the data and cover a broad range of motion, (Li et al., 2021; Kucherenko et al., 2021b;a) uses a motion-specific code space that represents motion attributes.

All of the state-of-the-art models mentioned above generate gestures frame-by-frame. We suggest that this architecture drains model capabilities on local dependencies at the expense of global features and diversity. We therefore reformulate the problem as a machine translation task. Inspired by VQ-VAE, we propose a method that combines VQ-VAE, Denoising Autoencoders and weakened decoders to learn a discretized latent space. Finally, we use an autoregressive model on the quantized motions to produce co-speech gestures from the input.

## 3 METHOD

We considered Yoon et al. (2019) as our baseline since we also focus on word embedding textual features (Bojanowski et al., 2017) as the control signal for gesture generation. The authors in (Yoon et al., 2019) used an RNN based Encoder-Decoder architecture with a soft attention mechanism (Bahdanau et al., 2014; Cho et al., 2014). The encoder extracts textual features, and the decoder produces co-speech gestures frame-by-frame. We extend the baseline model by utilizing representation learning approaches. Our proposed system has the following steps.

- Pose representation learning at the frame-level
- Discrete motion representation learning at the sequence-level.
- Translation from text to the learned discrete motion representation space.

Learning powerful representations without supervision is of utmost importance to reduce problem complexity. Autoencoders are unsupervised models that learn significant data features and discard spurious patterns by minimizing the reconstruction error (Kingma & Welling, 2013). They consist of an encoder network followed by a decoder network. An autoencoder's bottleneck finds a shared data representation (Goodfellow et al., 2009; Bengio et al., 2007) by learning the correlations between input dimensions and reconstructs them from a low-dimensionality representation. We used two different autoencoders with different components, both at the frame-level and sequence-level, as explained in the following sections.

### 3.1 FRAME-LEVEL AUTOENCODER

Inspired from Kucherenko et al. (2019; 2021a) we used a Denoising Autoencoder (DAE) architecture (Vincent et al., 2010; Goodfellow et al., 2016) to reduce dimensionality at the frame level.

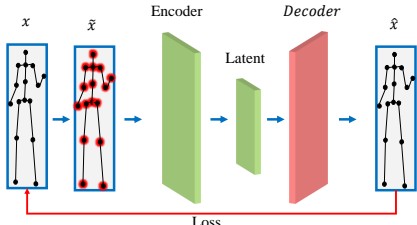

Figure 1: Pose DAE: Representation learning at the frame-level.

DAE learns a lower-dimensionality representation of data while corrupting the data intentionally through additive isotropic Gaussian noise. The bottleneck in the middle of DAE's encoder and decoder is generated from the noisy input, and the DAE learns to reconstruct the original input from it. DAEs can capture fundamental structures in the input distribution while preventing it from simply learning the identity function as each feature is encoded and decoded independently to the others (Vincent et al., 2010). The frame-level DAE includes an encoder and decoder as follows. The $DAE$ $Encoder$ maps a noise injected input $\widetilde{m}_f$ from pose space to the representation $z_f$ and decodes the representation $z_f$ back to a single frame $\hat{m}_f$ in motion space.

$$z_f = Encoder_{DAE}(\widetilde{m}_f) \; ; \; \widetilde{m}_f = m_f + N(0, I) \tag{1}$$

$$\hat{m} = Decoder_{DAE}(z_f) \tag{2}$$

As the DAE minimizes the reconstruction mean square error (MSE) loss, it learns a more informative representation to resemble the original input closely. Figure. 1 illustrates our representation learning for poses at the frame level.

## 3.2 SEQUENCE-LEVEL AUTOENCODER

In this step, we aim to create a vocabulary over the gesture space in which we will have a specific token for a set of similar motion sequences in the real world. We use a variational autoencoder (VAE) framework armed with a discrete latent representation (Oord et al., 2017). Given a set of observations, a Vector Quantized Variational Autoencoder (VQ-VAE) learns to create a motion vocabulary by parameterizing the posterior distribution of discrete latents.

In more detail, the VAE defines the posterior distribution as $p(z|x) \propto p(x|z)p(z)$. Typically, the prior $p(z)$ has been considered $z \sim N(o; I)$ on the latent variable $z \in \mathbb{R}^D$ where $D$ is the bottleneck's dimensionality. Accordingly, the VAE encoder creates a posterior distribution $q(z|x)$ over the latent representation of input $x$. Meanwhile, for a chosen approximate posterior $q(z|x)$, the decoder is trained on the reparametrized sample $\widetilde{z} \sim q(z|x)$ instead of deterministic encoded $z$ (Kingma & Welling, 2013). The VAE arranges the latent space such that motions with similar movements are projected close to each other while reducing the reconstruction loss (Kingma & Welling, 2013; Stewart et al., 2021).

Additionally, we consolidated a denoising characteristic to the variational autoencoder framework (DVAE), presenting a more flexible and robust posterior distribution approximation than the standard VAE (Im Im et al., 2017). It can be considered as a standard VAE with the denoising criterion, which samples a noise injected input $\hat{x} \sim p(\hat{x}|x)$ rather than $x$ itself. Afterward, it samples $\widetilde{z} \sim q(z|\hat{x})$ using an encoder network, and samples the reconstructed input from the $p(x|z)$.

Furthermore, we discretize the latent space by decomposing it into a set of embedding vectors (Oord et al., 2017). Previous work has shown that VQ led to better representation learning, prevented mode collapse, and provided high reconstruction resolution (Oord et al., 2017; Razavi et al., 2019; Chorowski et al., 2019; Baevski et al., 2019). Additionally, discretization allows employing algorithms from the NLP community to model long-range temporal dependencies rather than imperceptible details at the frame level.

In this model, shown in Figure 2, the posterior $q(z|\hat{x})$ and prior $p(z)$ distributions are categorical. Indeed, samples from these distributions map to tokens from a codebook of $K$ embedding vectors $e_i \in \mathbb{R}^D, i \in 1, 2, ..., K$. The discrete latent variables $z^{\ddagger}$ is determined from the continuous VAE's encoder output $z$. VQ-DVAE finds the $z$ nearest neighbour embedding vector $e_i$ in the $K$ embedding vectors of $\mathbb{R}^D$. As shown in equation 3, the posterior categorical distribution $q^{\ddagger}(z|x)$

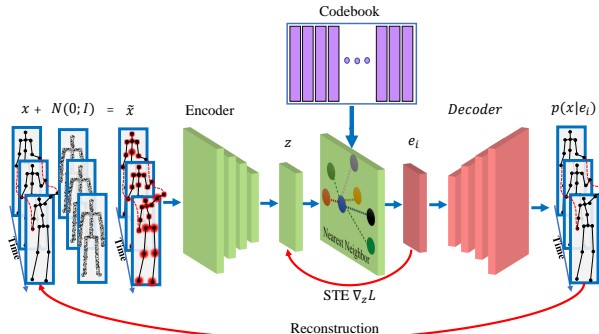

Figure 2: Motion VQ-DVAE: Discretized representation learning at the sequence-level.

probabilities represent a one-hot. Afterward, the corresponding embedding $e_i$ is fed to the decoder for the reconstruction process.

$$q^{\ddagger}(z^{\ddagger} = k|x) = \begin{cases} 1 & \text{for } k \arg\min_j \|z - e_j\| \\ 0 & \text{Otherwise} \end{cases} \tag{3}$$

Despite the standard VAE, the prior is a uniform distribution over the $K$ elements in the codebook. Therefore, the KL-divergence term usually incorporated into the ELBO is constant and safely removed. equation 4 defines the loss term for the VQ-DVAE from Oord et al. (2017). Since equation 3 is not differentiable, similar to the straight-through estimator (Bengio et al., 2013), we consider gradients from decoder inputs $e_i$ for encoder outputs $z$.

$$\mathcal{L} = \log p(x|z^{\ddagger}) + \|sg[z] - e\|_2^2 2 + \beta\|z - sg[e]\|_2^2; \tag{4}$$

The first term stands for the reconstruction loss and trains the encoder and decoder. Although the velocity does not appear on the loss function explicitly, we feed the input concatenated with its derivative as input to the VQ-DVAE. In fact, we encourage the model to reconstruct not only the input sequence frame by frame, but also its derivative. This can be counted as a proxy for motion dynamics (Yoon et al., 2019). The second term encourages the embeddings to move closer to the encoder output. On the other hand, the third loss term, named commitment loss, encourages the encoder to generate latents close to the assigned embeddings from equation 3. Meanwhile, it prevents the volume of the embedding space from growing arbitrarily.

Note that sg means "stop-gradient", so we do not propagate the gradient w.r.t. that term. Therefore, the first term updates both the encoder and decoder, the second term updates the codebook embeddings, and the third term updates the encoder to commit to its embedding.

### 3.3 TRANSLATION FROM TEXT TO GESTURE VOCABULARY

Finally, having a gesture vocabulary, the co-speech gesture generation resembles a machine translation task of English to the Gesture domain. Consequently, we apply a purely autoregressive model of sequence-to-sequence known as a machine translation model. For instance, n-gram models (Jurafsky & Martin, 2009), convolutional neural language models (Dauphin et al., 2017; Bai et al., 2018) attention-based models (Vaswani et al., 2017; Wang et al., 2019), etc. yielded attractive results. We use a two-layered bidirectional gated recurrent neural network (GRU) (Cho et al., 2014), with a soft attention mechanism (Bahdanau et al., 2014). After the translation task of input text to gesture tokens is complete, we use the VQ-DVAE's decoder to reproduce the entire gesture sequence followed by further post-processes such as the Savitzky-Golay smoothing filter (Savitzky & Golay, 1964).

## 4 EXPERIMENT

This section describes our implementation and experiment details, such as the specifics of the training data, model, and the training techniques we used. We obtained our training set from the Trinity dataset (Ferstl & McDonnell, 2018), which contains 23 clips, each approximately 10 minutes in length, with 244 minutes of aligned speech text, audio, and gesture training data in total. Using the GENEA challenge train and test set (Kucherenko et al., 2020b) also enables us to benchmark our results over provided test cases, including baselines and submitted models. Since this research

focuses on gaining the most semantic information from text inputs, we do not use the provided audio data in our approach.

The dataset is captured with a 53 marker setup and 20 Vicon cameras at 59.95 frames per second (FPS). The motion data was stored as a time series of Euler rotations for each joint in the BioVision Hierarchy format (BVH) with 59.95 frames per second. Since we focus on the upper body, we obtain the corresponding 15 upper-body joints out of the available 69 body joints. Moreover, Euler angles were converted to joint positions in 3D space and normalized regarding shoulder length. We also down-sampled gestures to the frame rate of 20 FPS and removed finger motions due to the low accuracy of recordings. Although the dataset provides aligned word transcription in JSON format, it is inaccurate and might mislead the model. We hence applied the Gentle forced aligner algorithm (Ochshorn & Hawkins, 2017) to obtain exact timing information for each word.

## 4.1 DAE

We train our system on 15 joints from the upper body, including: Spine, Spine1, Spine2, Spine3, Neck, Neck1, Head, RightShoulder, RightArm, RightForeArm, RightHand, LeftShoulder, LeftArm, LeftForeArm, and LeftHand. Furthermore, we used 3x3 rotational matrices instead of 3D coordinates of each 15 joints to train the DAE network. Thus, the size of input and output vectors for pose representation at frame-level was 3x3x15=135. We also standardized each dimension to a mean of zero and a maximum absolute value of one for fast convergence in training.

The DAE consists of one linear layer of input size to the bottleneck dimensionality followed by a Tanh activation as an encoder and a linear layer of bottleneck to input size as a decoder and learning rate of 0.001. We injected Gaussian noise of standard deviation 0.1 to input data and the dropout rate was set to 0.2. To decide proper dimensionality for the bottleneck, we trained the model with different hidden dimensions fo 20 epochs. Similar to Thangthai et al. (2021); Kucherenko et al. (2019), we picked 40 dimensions to represent every single motion frame.

## 4.2 VQ-DVAE

In this section, we explain further details of the VQ-DVAE model as well as the training process. At this stage, we aimed to map a sequence of $s_i$ to a latent representation. We selected sequences with length of 30 frames and a stride size of 10 frames. Also, considering the joint velocity as a proxy of motion dynamics (Yoon et al., 2019; Kucherenko et al., 2020a; 2021a), we concatenated the derivative of poses representations to the input . Therefore, we feed 40+40 = 80 features per frame to the autoencoder to encode and reproduce it. VQ-DVAE consists of an encoder, a quantized latent space embeddings as bottleneck, and a decoder. In order to learn sequential data, by stacking Bi-directional GRU networks on top of each other, we defined a multi-layer bi-directional recurrent neural network architecture for both encoder and decoder networks with the hidden size of 200. Consequently, we compress $80 * 30 = 2400$ dimensions to the continuous latent space of $200 * 2 = 400$ dimensions. Afterward, we quantize this variable $z$ into the nearest neighbour embedding $e_i \in \mathbb{R}^D, D = 400, i \in 1, 2, ..., K$ and feed it to the decoder as discussed before. We empirically chose the number of embedding vectors $k$ equal to 300.

## 4.3 WEAKENED DECODERS

As mentioned earlier, VAEs suffer from posterior collapse when a strong decoder network is used. One simple yet effective solution to this problem could be weakened decoders (Yang et al., 2017; Semeniuta et al., 2017). A conditional RNN decoder receives the last generated output frame as input to the current step. In our problem, continuity, short-range correlation, is a strong assumption (Alexanderson et al., 2020; Srivastava et al., 2015). For instance, the joint positions of a frame are the same as the previous one with subtle changes. A conditioned decoder simply determines these correlations and neglects extremely subtle movements requiring long-term dependencies (Srivastava et al., 2015). An unconditioned decoder that does not receive that input enforces the encoder to find this information and put it into the encoded vector. To learn a more robust and informative representation, we weaken the decoder by freezing its weights while minimizing the ELBO loss. Training the model with no gradient affecting the decoder enforces the encoder to learn informative and robust representation. In this setup, the decoder only propagates the encoder output through time, and

the encoder carries reconstruction loss. However, after 20 epochs, we also trained the decoder in a conditioned fashion for another 20 epochs concerning smooth reproducibility at inference time.

## 5 EVALUATION

In this section, we elucidate our comprehensive experiments to validate the effectiveness of the proposed method. We present an ablation study in the Appendix to substantiate the effectiveness of all the components of VQ-DVAE. We describe the baselines systems as well as the objective and subjective measures we used to assess our method.

### 5.1 BASELINE SYSTEMS

We assessed our system against the ground-truth, and three recently published co-speech gesture generation systems. Our ground-truth was taken from the natural motions of the actor for a given speech segment. Baseline Text (BT) from Yoon et al. (2019) is the most similar work to ours since we both used Fasttext word vectors (Bojanowski et al., 2017) as our features from the input text transcript. Also, we both used the Bahaduna RNN Encoder-Decoder architecture with a soft attention mechanism (Bahdanau et al., 2014; Cho et al., 2014). The encoder extracts text features, and the decoder generates poses frame-by-frame. However, our sequence-to-sequence model maps extracted features from the text into a series of motion symbols, with each symbol representing 30 frames of motion. Baseline BA proposed by Kucherenko et al. (2019) used audio as input to generate co-speech gestures composed of an encoder-decoder structure. In this model, the encoder maps audio input to a sequence of learned pose representations while the decoder projects them back to poses. The third baseline model (BTA) (Korzun et al., 2020) achieved impressive results (Kucherenko et al., 2020b) using both audio and text modalities. In order to extract acoustic and textual features, BTA combined two separate encoders, one for each modality. This work was interesting since it used both modalities and scored the highest in human-likeness and second-highest in terms of appropriateness among (Yoon et al., 2019; Kucherenko et al., 2019; Alexanderson et al., 2020; Lu et al., 2021; Thangthai et al., 2021).

### 5.2 OBJECTIVE MEASURES

Although improving subjective measures is our ultimate goal, they are costly, time-consuming, and require human labour. Moreover, evaluating generative models is tricky since there is not a one-and-only true motion sequence for a given speech utterance. Accordingly, we support our experimental results with numerical evaluations recently proposed to unveil gesture qualities.

Average jerk and velocity metrics have been used in prior work to quantitatively assess gesture systems (Kucherenko et al., 2020b). We used average jerk and velocity metrics proposed by Kucherenko et al. (2019). Jerk, the third derivative of joint positions, characterizes smoothness by calculating the rate of acceleration in a movement. We used the average over third derivative of joints as an objective metric to compare systems. We also posit that plausible generated gestures should follow similar velocity characteristics to the ground truth. Note that the gestures were converted from joint rotational angles to 3D joint positions. Hellinger distance (Hellinger, 1909) has extensively been used to compare two distributions. Accordingly, we compare each system to the ground truth to see how close the gestures are to the natural motion w.r.t that structural aspect.

Additionally, we introduce an objective metric to evaluate gesture diversity over the long term, thanks to our proposed discretized latent space at the sequence level. Indeed, we used our trained VQ-DVAE model to cluster predicted gestures for each condition at the sequence level. We adopted the Hellinger metric to evaluate how a system's output distribution follows the real-world gesture vocabulary distribution. The Hellinger distance metric was applied between each system and the ground truth to evaluate its closeness in terms of diversity.

### 5.3 SUBJECTIVE MEASURES

Our ultimate purpose is to generate realistic gestures that are as valid and natural as the ground truth. Therefore, we conducted a human study to evaluate our performance subjectively. Comparable to

Table 1: Objective Evaluation Results

| System | Average Jerk | Hellinger Distance | | |
|---|---|---|---|---|
| | | Velocity | Acceleration | Diversity |
| GT | $1588.63 \pm 2899.61$ | 0 | 0 | 0 |
| BT | $795.09 \pm 73.27$ | 0.0988 | 0.1651 | 0.7495 |
| BA | $1275.43 \pm 78.21$ | 0.0793 | 0.1592 | 0.6298 |
| BTA | $904.19 \pm 81.61$ | 0.0710 | 0.1268 | **0.5715** |
| **Proposed** | $\mathbf{1601.22 \pm 104.03}$ | **0.0511** | **0.06465** | 0.59184 |

Table 2: Subjective Evaluation Results

| System | Human-Likeness | | Appropriateness | | Diversity | |
|---|---|---|---|---|---|---|
| | Mean | STD | Mean | STD | Mean | STD |
| GT | 0.69 | 0.22 | 0.71 | 0.23 | 0.72 | 0.21 |
| BT | 0.41 | 0.23 | 0.31 | 0.21 | 0.43 | 0.20 |
| BA | 0.39 | 0.20 | 0.42 | 0.21 | 0.45 | 0.19 |
| BTA | **0.50** | 0.22 | **0.53** | 0.22 | 0.45 | 0.20 |
| **Proposed** | **0.50** | 0.20 | 0.45 | 0.22 | **0.53** | 0.18 |

the prior work in this area (Ginosar et al., 2019; Salvi et al., 2009; Kucherenko et al., 2020b; Alexanderson et al., 2020; Thangthai et al., 2021; Lu et al., 2021; Yoon et al., 2019; Korzun et al., 2020; Kucherenko et al., 2019), we evaluated generated gestures regarding perceived *human-likeness* and *appropriateness* of the motion given the speech context. These two measures indeed disentangle motion quality from the relevancy to the speech context. In addition, we also assessed systems in terms of diversity as opposed to repetitive movements over longer intervals. Specifically, we asked the following questions in our human-study: "How human-like does the motion appear?", "How appropriate are the gestures for the speech?" and "How diverse does the motion appear?".

For the appropriateness evaluation, we selected 40 speech segments each contains a complete sentence or phrase with an average of 10 seconds. Similarly, we selected 40 muted segments randomly for the human-likeness study and diversity study with a fixed length of 10 seconds and 20 seconds, respectively.

## 5.4 EXPERIMENTAL PROCEDURE

We recruited participants from the Amazon Mechanical Turk (MTurk) to evaluate the results subjectively. MTurk is a crowdsourcing website proven effective in recruiting more diverse participants than in college experiments (Keith et al., 2017). MTurk provides the option to set requirements for the intended population based on different factors, i.e. age, gender, nationality, HIT approval rate. The HIT approval rate is defined as the percentage of completed work that other requestors have approved for that specific person. We posted a Human Intelligence Task (HIT) on MTurk containing an external link to our web interface. More details about our human-study interface are provided in the Appendix section. Participants were asked to complete the HIT, obtain a survey completion code, and enter it in MTurk to be compensated based on the minimum wage per hour in their country. We restricted our intended population to collect high-quality results by setting the minimum HIT approval rate to $95\%$ and approved HITs greater than 5000. We also required participants to be located in Canada due to ethical review board policies.

## 6 RESULTS

We recruited 24 participants and filtered out 6 judgments based on attention checks and ratings to the ground-truth gestures as a superior system over all conditions. The average age was 33.1 (STD=8.9) years with 12 men and 6 women, all English native speakers living in Canada. Among accepted judgments, the average experiment duration was approximately 41 minutes. The mean and standard deviation (STD) of ratings for the three studies are presented in Table 2.

Objective results on the 20 minute test set for all conditions is summarized in Table 3. We report the mean and standard deviation of the average jerk. Table 1 also presents the Hellinger distance of velocity and acceleration histograms to the ground truth. The diversity column shows the Hellinger distance on vocabulary frequency based upon assigned labels to constituent sequences (30 frames) obtained from VQ-DVAE.

## 7 DISCUSSION

Gesture generation is challenging, especially in terms of appropriateness for the speech. Consequently, the difference between systems is subtle and a long way from the natural gestures. Objective metrics are consistent with the subjective results; however, we can see that different systems performed differently on evaluation metrics.

The subjective study shows that our model is preferred over all the systems in terms of human likeness. This is also aligned with similarity metrics in objective evaluation, where the proposed method

ranked first in fundamental factors, i.e. jerk, acceleration, and velocity. Indeed, we can conclude that our representation learning method efficiently captured essential features and reproduced more natural gestures. Prior research has also shown that vector quantization led to more diverse and high-quality outputs. The BTA is the best in terms of appropriateness and our method is ranked second. We only used text features similar to the BT, while BTA used more advanced semantic features alongside acoustic features. We interpret it as the effectiveness of quantization against continuous regression resulted in better performance. Furthermore, raters perceived our model outputs as more diverse, showing our model's capability to overcome the mode collapse problem. As shown in Table 3, the proposed gesture generation method is significantly closer to the ground truth than the BT and approximately similar to BA and BTA in terms of the appearance of gesture sequences distribution.

Subjective and objective results suggest that we avoided the average gesture problem and generated more diverse and natural outputs than repetitive gestures. Vector quantization of motion representation narrowed the gesture space complication down to a set of vocabulary in a codebook that clusters similar sequences into a specific motion symbol. Although we significantly reduced the dimensionality with discrete encoding, both objective and subjective measurements show that generated gestures maintain a high quality in terms of naturalness. Quantization of motion representation space promotes the gesture generation model to focus on longer dependencies rather than local correlation. Consequently, we can apply machine translation algorithms from the NLP community to perform co-speech gesture generation tasks using entropy family loss instead of regression type.

We found that weakening the decoder strongly impacts the learned representation during the training process, resulting in better clusterings. However, its reproducibility was not smooth, and we later trained the decoder for 20 epochs while other parameters were fixed.

### 7.1 Limitations and Future Work

The first limitation of our work was that the dataset used in this research was limited to one person speaking in a monologue situation. Different persons have their own gesticulating style and different environments lead to complex speech and gestures. In future work we should consider datasets with mixed speakers in different environments. Another limitation of our research is that we trained our model on uppe,r body while fingers were excluded. Lower body is also important e.g. stepping forward and backward motions, standing still, approaching, facial expressions and fingers motions.

Although our system showed a significant improvement over baselines, it is still far behind human-generated motions. Kucherenko et al. (2020b) reported that raters were inclined to rate mismatched speech gestures from the ground truth, higher than synthesized gestures for a given input. Therefore, it is possible that we achieved a high appropriateness rate, especially compared to the Baseline Text (BT), due to the higher human-likeness quality of our system instead of appropriateness.

In this study, we did not involve audio and only used speech text. Punctuations such as question marks were also not provided in the source corpus. Therefore, generated gestures may lack movements relevant to acoustic features such intonation at the end of a sentence to indicate question (Pouw et al., 2020). Current work can be improved by adding punctuation using an automatic punctuation restoration system (Courtland et al., 2020). The proposed method can be extended by looking for correspondences between motions and audio features.

## 8 Conclusion

We proposed a fully unsupervised co-speech gesture generation system that combines representation learning methods and a machine translation algorithm. To the best of our knowledge, this is the first method that uses a representation learning algorithm at the sequence-level for the text-to-gesture generation task and provides a new state-of-the-art baseline in in this area. We connected a pose (frame-level) representation learning method and a quantized motion (sequence-level) representation learning, each trained separately. We also introduced a new objective evaluation metric that calculates the Hellinger distance of motions occurrence distributions as a measure for diversity. Finally, we trained a machine translation model to translate English sentences to gesture vocabulary. We found that the discretized motion space causes the model to focus on longer dependencies and generate more diverse and human-like gestures. The main limitation of our work was that we did not include audio features. In future work, we will consider more advance semantic features, audio fea-

tures as well as emotional features, i.e. valence and arousal (Lim et al., 2011). The dataset was also not large enough to generalize a high-performance model, especially in terms of appropriateness. This can be addressed by creating a proper dataset through an automated process from currently available datasets. Furthermore, we will investigate hierarchical vector quantization (Razavi et al., 2019), to generate more natural and diverse gestures.

## 9 REPRODUCIBILITY

Towards reproducibility of our results, we include here a link to our code implementing the proposed model as well as statistical analysis used in this paper:

`https://osf.io/xznb3/?view_only=8ba7c43f839242678e89f811e4763b6b`

We also provide the following link to visualizations and the human-study interface described in our Experiments section:

`https://osf.io/65q4p/?view_only=aa54d9fea4f1452a844a6acf6f6f2ac4`

Finally, the Appendix includes more details regarding the human study interface, in order for others to reproduce the subjective results.

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

# A APPENDIX

## A.1 INTERFACE

The human-study interface, shown in Fig. 3, was implemented in Unity3D (Haas, 2014) and published in a WebGL format. We used Blender software (Community, 2018) to convert BioVision Hierarchy (BVH) files to a Filmbox (fbx) format that Unity3D can import as a humanoid animation. The avatar has 15 joints, excluding fingers in order to be consistent with our benchmarks.

The landing page of the experiment interface presented a consent form and instructions on how to use the interface with a quick guide tour. We randomly picked 10 clips from each of the three aforementioned segment pools to keep the experiment within 30 minutes and avoid exhausting raters. The first 10 clips shown to participants were from the human-likeness study. Next, raters were asked to use their headphone and test the audio settings to rate appropriateness stimuli. In the end, they were asked to rate 10 muted long clips from the diversity pool. To evaluate several systems simultaneously, we used a methodology similar to the (Kucherenko et al., 2020b) inspired by the Multiple Stimuli with Hidden Reference and Anchor test (Series, 2014). We used a page-wise strategy such that on each page, we included motions from all conditions corresponding to a specific segment assigned to that page. Therefore, we could employ pairwise statistical tests since stimuli were rated in parallel. We selected the 100-point rating scale and labelled them: "Bad", "Poor", "Fair", and "Excellent" within intervals of 20 points (Kucherenko et al., 2020b). Meantime, we randomized the order of clips on each page. After rating 30 stimuli, participants were asked to complete a demographic questionnaire. To see if a participant was paying enough attention and eliminate inattentive raters, we included an attention check on each page where we asked the participant to rate a specific value for a stimulus. On each page, the attention video and corresponding answer were selected randomly. Afterward, we excluded collected data with more than four failures in attention checks.

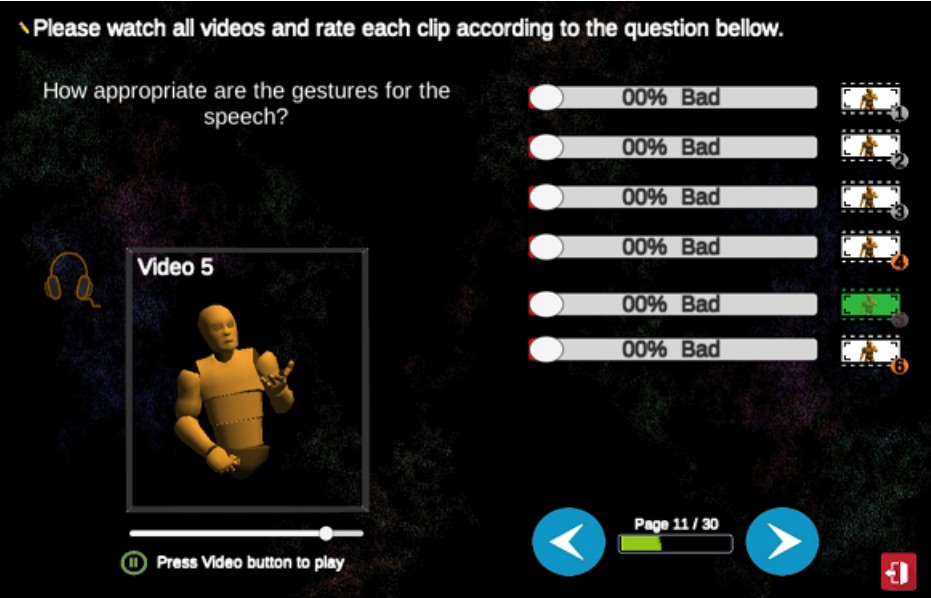

Figure 3: A screenshot of a page with stimuli from the human-study interface.

## A.2 ABLATION STUDY

To evaluate the essence of the contributions of each component in our method, we assess the latent representation space as well as the final output within an ablation study. To evaluate the latent representation, we analyze how two shifted sequences related to each other in latent space. We defined a Neighbour Sample Distance (NSD) metric, which measures the average distance between a sequence and its shifted version. We assume that in a good representation space, shifted samples' distances is smaller concerning the average distance in that space. We also apply the Fréchet gesture distance (FGD) Yoon et al. (2020) to compare distributions on the latent gesture space between

Table 3: Ablation Study Results

| System | W | D | F | C | Der | VAE | VQ-VAE | Latent distance | | | FGD | Wasserstein |
|---|---|---|---|---|---|---|---|---|---|---|---|---|
| | | | | | | | | *10 Frames* | *20 Frames* | *All* | | |
| Proposed | 30 | ✓ | ✓ | ✗ | ✓ | ✗ | ✓ | $0.50 \pm 0.16$ | $0.61 \pm 0.16$ | 24.64 | 4.83 | 18.69 |
| Proposed - W | 20 | ✓ | ✓ | ✗ | ✓ | ✗ | ✓ | $N.A$ | $N.A$ | $N.A$ | 7.95 | 22.36 |
| Proposed - W | 15 | ✓ | ✓ | ✗ | ✓ | ✗ | ✓ | $N.A$ | $N.A$ | $N.A$ | 6.62 | 37.02 |
| Proposed - W | 10 | ✓ | ✓ | ✗ | ✓ | ✗ | ✓ | $N.A$ | $N.A$ | $N.A$ | 7.84 | 18.31 |
| Proposed - D | 30 | ✗ | ✓ | ✗ | ✓ | ✗ | ✓ | $0.53 \pm 0.16$ | $0.63 \pm 0.15$ | 24.49 | 4.57 | 15.58 |
| Proposed - F | 30 | ✓ | ✗ | ✗ | ✓ | ✗ | ✓ | $0.35 \pm 0.11$ | $0.50 \pm 0.13$ | 24.39 | 7.71 | 22.12 |
| Proposed - C | 30 | ✓ | ✓ | ✓ | ✓ | ✗ | ✓ | $0.53 \pm 0.14$ | $0.68 \pm 0.16$ | 23.41 | 4.68 | 16.61 |
| Proposed - Der | 30 | ✓ | ✓ | ✗ | ✗ | ✗ | ✓ | $0.51 \pm 0.15$ | $0.62 \pm 0.15$ | 25.52 | 4.75 | 22.10 |
| Proposed - VQ | 30 | ✓ | ✓ | ✗ | ✓ | ✗ | ✓ | $0.75 \pm 0.26$ | $0.89 \pm 0.24$ | 43.65 | 5.23 | 22.13 |
| Proposed - Vanilla | 30 | ✗ | ✗ | ✗ | ✗ | ✗ | ✗ | $0.73 \pm 0.20$ | $0.78 \pm 0.17$ | 5.56 | 19.79 | 109.95 |
| Proposed - D&Der | 30 | ✗ | ✗ | ✓ | ✗ | ✗ | ✗ | $0.39 \pm 0.14$ | $0.53 \pm 0.15$ | 25.4 | 4.55 | 20.10 |

real and generated gestures. The more generated motions similar to the ground truth on the latent distribution, the smaller FGD value is.

