# OpenReview forum: "Gesture2Vec: Clustering Gestures using  Representation Learning Methods for Co-speech Gesture Generation"
_ICLR.cc/2022/Conference — ICLR 2022 Submitted_

### Official Review · Reviewer_aN3z · 2021-10-26

**Correctness:** 3
**Technical Novelty And Significance:** 1
**Empirical Novelty And Significance:** 2
**Recommendation:** 5
**Confidence:** 4

**Details Of Ethics Concerns:**

I do not have a concern. It is stated in the paper that the authors were required participants to be located in Canada due to ethical review board policies.

**Main Review:**

Strengths:
- I appreciate that the related work section which discusses each prior art very well with their cons and pros.
- The proposed method brings some improvements over the baseline methods for metric jerk, velocity and accelerarion when tested on Trinity dataset.

Weaknesses:
-  The technical novelty of the proposed method is questionable. It is important to highlight that this is not the first time variational autoencoders are adapted for this task. I found the proposed method incremental, such as adding the denoising property to the existing Vector Quantized Denoising Variational Autoencoder architecture.
- The paper should include ablation study showing the effectiveness of the components compared to the other possibilities. For example, the authors claim that “….we consolidated a denoising characteristic to the variational autoencoder framework (DVAE), presenting a more flexible and robust posterior distribution approximation than the standard VAE”. However the correctness of this argument was not empirically justified.
 - I found the experimental analysis limited. The proposed method should be validated also on other datasets and following that, it should be compared with other state of the arts too (e.g., "MoGlow: Probabilistic and Controllable Motion Synthesis Using Normalising Flows, ACM Trans. Graph., code available").
I am aware that prior art, for example, Yoon et al. 2019 was used TED gesture dataset, which is a larger scale dataset compared to Trinity. Another dataset is Gesture-speech dataset collected by Takeuchi et al. Gesture-speech dataset include audio in which one can obtain the transcriptions. Authors should include these datasets into their analysis, to show that the findings generalize across various conditions.
- It is also an important limitation that the proposed method was tested on a dataset having only monologues. The interaction scenarios is perhaps much more challenging and also more realistic, assuming that such a system can be deployed to social robots which interact with other social agents.
- “Although our system showed a significant improvement over baseline….” this argument should be justified by a significance test.
-  The applied human study should be better described in the main paper, it is not clear to me what is the reliability score of the human annotations.
- Figure captions require a better explanation. E.g., in Fig 1, x, x\tilde, x\head should be described as well as Loss should be precisely mentioned as MSE Loss. The similar comment applies for Fig.2.
- My understating is that the proposed method is not end-to-end. Can you confirm? If so, this fact should be included to the discussions as some of the baseline methods are end-to-end.

**Summary Of The Paper:**

This paper tackles with co-speech gesture generation. First a denoising autoencoder is applied to the body pose detected at each frame. Then, a Vector Quantized Denoising Variational Autoencoder learns to create a motion vocabulary by parameterizing the posterior distribution of discrete latents. As the third step,  a two-layered bidirectional gated recurrent neural network with a soft attention mechanism is used for translation from text to gesture vocabulary. The decoder of Vector Quantized Denoising Variational Autoencoder is used to reproduce the entire gesture sequence.

**Summary Of The Review:**

- limited technical novelty
- limited experimental analysis

---

> ### Comment · Reviewer_aN3z · 2021-11-30
> **final decision**
>
> I have read all reviews. It looks like Reviewer mN3D raised a lot of questions regarding implementation details. I had criticism on technical novelty and the limited experimental analysis. I do not see any reply from authors. I, therefore, lean towards a reject.

---

### Official Review · Reviewer_RQVL · 2021-11-01

**Correctness:** 3
**Technical Novelty And Significance:** 3
**Empirical Novelty And Significance:** 3
**Recommendation:** 6
**Confidence:** 5

**Main Review:**

### Strengths
1. The idea of representing gesture sequences as vector-quantized features is technically sound and clearly presented.
2. The experimental results show reasonable benefits over prior work, especially in objective evaluation metrics.

### Weaknesses
1. Related work appears incomplete. There is at least one other work on generating gestures from input text transcripts:
Bhattacharya, U., Rewkowski, N., Banerjee, A., Guhan, P., Bera, A., & Manocha, D. (2021, March). Text2Gestures: A Transformer-Based Network for Generating Emotive Body Gestures for Virtual Agents. In 2021 IEEE Virtual Reality and 3D User Interfaces (VR) (pp. 1-10). IEEE.

2. Some technical aspects of the proposed method are unclear to me:
    1. For the DAE (Section 4.1), how do the authors ensure that the decoder outputs are valid rotation matrices?
    2. What reconstruction loss is used for the DVAE that works with gesture sequences? Is it a mean squared error (MSE) similar to that used by the frame-level DAE? Since MSE forces the mean values of the poses in the input and the output gestures to be close, it may cause the outputs to be temporally smoothed out. If that is the case, then how does the proposed method provide the largest average jerk (Table 1, last row)? If not, do the authors perform any additional steps to safeguard against the smoothing?
    3. Yoon et al. 2020, as referred to by the authors, have already proposed an objective measure for gesture diversity, dubbed Fréchet Gesture Distance (FGD). Since the authors propose their own metric, the Hellinger distance, for the same purpose, it would have been appropriate to have at least some discussion on how the two metrics compare. More importantly, what is the motivation to propose a new metric over the existing one?
    4. Before the user study, were the users shown any good and bad examples (or something similar), as determined by the authors, for the questions they were subsequently asked?
    5. Did the authors perform any inter-rater agreement, e.g., using Fleiss' Kappa measure, to verify the reliability of the responses?
3. In Section 7.1, the authors mention that they were limited to one person in their work. However, multi-person datasets are already available, such as the TED Gesture Dataset proposed by Yoon et al. 2019, and used by both Yoon et al. 2019 and Yoon et al. 2020. Are there any specific reasons why the authors did not work with this dataset? Moreover, in Section 4 (page 5, last line), the authors mention that they only focus on upper body gestures. Again, were they any specific reasons to eliminate the lower-body gestures from the training and the testing processes?

**Summary Of The Paper:**

The authors present a method to generate co-speech gestures by learning a vector-quantized representation space of gestures coupled with a machine translation from natural language sentences to that representation space. They design a denoising variational autoencoder (DVAE) to learn the encoding space corresponding to gesture sequences, and then quantize that encoding space using nearest neighbor embeddings. Once the vector quantized representations are learned, they use a sequence-to-sequence network to translate the text transcript of the input speech to those representations, which they subsequently decode into the output co-speech gestures. They train and test their method on the Trinity Gesture Dataset and show the improvements achieved by their method on both objective and subjective evaluations.

**Summary Of The Review:**

The proposed method is technically sound and attempts to solve the challenging problem of co-speech gesture generation using an insightful approach, that of vector-quantizing the latent representation space of gestures and mapping from natural language sentences to it. However, a number of technical details were unclear to me, which makes the overall method and evaluations hard to follow. I invite the authors to respond to my comments under "weaknesses" before I can strongly recommend the paper for acceptance.

---

### Official Review · Reviewer_mN3D · 2021-11-03

**Correctness:** 2
**Technical Novelty And Significance:** 2
**Empirical Novelty And Significance:** Not applicable
**Recommendation:** 3
**Confidence:** 5

**Main Review:**

### **Strengths**

- The choice of learning a representation of a gesture sequence instead of a single frame can indeed provide the model with the ability to generate plausible gesture sequences. This is corroborated by somewhat better subjective evaluation scores.

### **Weaknesses and Suggestions**
- The novelty of the contributions is overplayed in the paper. Most of the discussion is about prior work on VAE, VQVAE, crossmodal conditional VQVAE decoding [1]. Learning representations for gesture sequence is an incremental contribution (it is also quite similar to Hierarchical VQVAE[6]), but ablation studies discussing its importance is missing. Hence it is hard to judge whether these representations really helped with the performance.

- As the task is to generate a sequence of upper body gestures, it would have been more convincing to see some demo videos. While I appreciate the subjective evaluation studies, it is hard to judge the output of a generative model without actually having seen any examples.

- Method: The overall method explained in Sections 3.1 and 3.2 is effectively paraphrasing the VAE and VQVAE papers. Only a small portion on sequence level representation learning seems to be novel. It would have been better to see significantly more descriptions about the contributions instead of the prior work.

- A big chunk of the paper talks about the representation learning of co-speech gestures, yet there is no analysis of these representations. There are several challenges that could occur with a sequence level codebook, two of which are as follows:
  - How do two sequences, one of which is a shifted version of the other (i.e. the two sequences are x[0:30] and x[10:40]), appear in the latent space? Are they represented by the same code in the codebook? If yes, how does the decoder figure out where to start the gesture from, 0 or 10?
  - What is the optimal number of codes in the codebook that is able to represent the manifold of all possible gestures? How is the value of K=300 chosen? Often, if the value of K is higher than optimal, there can be codes that are very close to each other. This is similar to the challenge of choosing more cluster centers than the data requires. Here, what could happen is that the language encoder (in the third step) will now be uncertain about which code to choose from.

- Section 3.3 - In step 3, the text features are used to choose a sequence of codes which are later decoded into gesture sequences. An interesting metric here could be measuring the classification accuracy of these codes. This could also help measure the uncertainty of prediction of the gestures which is not possible in the system with implicit generative models like GANs.

- Section 4.2 VQ-DVAE - The derivatives (i.e speed/velocity) of the pose representations of the input were also fed in with the pose representations as a proxy for motion dynamics. This makes an assumption that transformations such as addition or subtraction in pose representations are equivalent to the same transformations in the original pose. This is in general not true as the network used here is non-linear.

- It is not clear when the decoder is frozen (Section 4.3 on Weakened decoders)? Is it during step 1, 2 or 3?

- Section 5.1 Baselines: Some of the baselines are missing including [2] (text + audio input with a gesture as output), [4](audio input and gesture as output) and [5] (text input and human body motion as output with a seq2seq architecture).

- Hellinger distance: The authors propose the use of Hellinger distance as an objective metric to measure the distance between the distributions of the generated gestures and ground truth gestures. While measuring the distribution distance is a good idea, it has been proposed for the same task in [2] and [3] with the use of Wasserstein Distance (i.e FID[2] or FGD[3]). There is a tradeoff between Hellinger distance and Wasserstein distance. They both have similar properties when the domains of the two distributions have some overlap. But as soon as the overlap between the distributions becomes zero, Hellinger distance fails to give any information on how far the two distributions are. Wasserstein distance on the other hand will still be able to give information on how far the two distributions are.

- Table 1: What is the average jerk of the ground truth gestures so high ($1588.63 \pm 2899.61$)? The interval covered by the GT jerk is [$-1310.98, 4400.24$] which encompasses the jerk vaues of all the models. Hence, this experiment can't definitively say much about the performance of baselines and the proposed model.

- Ablations: There are many hyperparameters in this approach, including codebook size K, gesture sequence length (which was taken as 30 arbitrarily), DAE bottleneck. In section 3.2, the description of the model indicates that it also predicts the velocity of the output poses. Why is that necessary? It was also not clear how that affects the learnt representations.

- The paper is fairly confusing, and hence unclear at times. The clarity of the paper can definitely be improved a lot with a significant re-write. For example:
  - In Section 4, it is mentioned that the Euler angles were converted to joint positions. But in Section 4.1, it is stated that 3x3 rotational matrices were used instead of 3D coordinates. These two statements are inconsistent.
  - In section 4.1, it is stated that the model is trained with differently sized latent embedding bottlenecks to find the optimal size. But the next statement mentions that a dimension size of 40 is chosen similar to Thangthai et. al. 2021 and Kucharenko et. al. 2019. These statements are contradictory.
  - It wasn't very clear how that learn representations of the DAE were used as part of the second VQ-DVAE. I had to infer this from the 40+40=80 dimensional vector in Section 4.2.
  - The terms DAE and VAE have been used interchangeably. It would be good to stick to one of them.


References
- [1] Richard, Alexander, et al. "MeshTalk: 3D Face Animation from Speech using Cross-Modality Disentanglement." ICCV 2021"
- [2] Ahuja, Chaitanya, et al. "No Gestures Left Behind: Learning Relationships between Spoken Language and Freeform Gestures." Proceedings of the 2020 Conference on Empirical Methods in Natural Language Processing: Findings. 2020.
- [3] Yoon, Youngwoo, et al. "Speech gesture generation from the trimodal context of text, audio, and speaker identity." ACM Transactions on Graphics (TOG) 39.6 (2020): 1-16.
- [4] Ginosar, Shiry, et al. "Learning individual styles of conversational gesture." Proceedings of the IEEE/CVF Conference on Computer Vision and Pattern Recognition. 2019.
- [5] Ahuja, Chaitanya, and Louis-Philippe Morency. "Language2pose: Natural language grounded pose forecasting." 2019 International Conference on 3D Vision (3DV). IEEE, 2019.
- [6] Razavi, Ali, Aaron van den Oord, and Oriol Vinyals. "Generating diverse high-fidelity images with vq-vae-2." Advances in neural information processing systems. 2019.

**Summary Of The Paper:**

In this paper, the authors propose an approach to perform co-speech gesture generation. A model Gesture2Vec is proposed to first learn the latent representation of gesture sequences. A vector quantized variational autoencoder is used to learn a codebook of different kinds of gesture sequences speaker's gesture manifold. Finally, the pre-trained decoder and codebook representations are conditioned on a sequence of language tokens to generate relevant co-speech gestures. This approach is tested on a public dataset containing a single speaker objective and subjective metrics.

**Summary Of The Review:**

While the motivation behind the learning representations of a gesture sequence instead of single frames is interesting, it is not executed very well. A lot of time has been spent explaining prior work instead of the contributions. The experiments to analyse the learnt representations is also lacking. Some of the details in the paper are confusing and hence needs significant revisions. And finally, being a generative modelling task, it would have been good to see some example videos.

---

### Decision · Program_Chairs · 2022-01-20

**Decision:**

Reject

**Comment:**

PAPER: This paper describes a method to generate visual gestures by learning an intermediate representation based on gesture sequences. This proposed method builds from previous work on VAE and vector quantized VAE.
DISCUSSION: The reviewers wrote some detailed reviews about the paper, bringing some valid concerns and asking some questions to the authors. Unfortunately, no responses were posted from the authors.
SUMMARY: After looking at all reviews, there was a general consensus among the reviewers that this paper was not ready for publication. We hope that the reviews will be helpful for the authors in revising their work for future submission.